# Interior Hotspot Engineering in Ag–Au Bimetallic Nanocomposites by In Situ Galvanic Replacement Reaction for Rapid and Sensitive Surface-Enhanced Raman Spectroscopy Detection

**DOI:** 10.3390/ijms231911741

**Published:** 2022-10-03

**Authors:** Iris Baffour Ansah, Soo Hyun Lee, ChaeWon Mun, Dong-Ho Kim, Sung-Gyu Park

**Affiliations:** 1Department of Nano-Bio Convergence, Korea Institute of Materials Science, 797 Changwondae-ro, Changwon 51508, Korea; 2Advanced Materials Engineering Division, University of Science and Technology (UST), 217 Gageong-ro, Yuseong-gu, Daejeon 34113, Korea

**Keywords:** surface-enhanced Raman spectroscopy, cellulose acetate paper, galvanic replacement reaction, silver-gold bimetallic nanocomposites, benzyl butyl phthalate

## Abstract

Engineering of interior hotspots provides a paradigm shift from traditional surface-enhanced Raman spectroscopy (SERS), in which the detection sensitivity depends on the positioning of adsorbed molecules. In the present work, we developed an Ag–Au bimetallic nanocomposite (SGBMNC) SERS platform with interior hotspots through facile chemical syntheses. Ag nanoparticles replaced by Au via the galvanic replacement reaction (GRR) provided hotspot regions inside the SGBMNC that remarkably enhanced the plasmonic activity compared to the conventional SERS platforms without the internal hotspots. The diffusion of analytes into the proposed interior hotspots during the GRR process enabled sensitive detections within 10 s. The SERS behaviors of the SGBMNC platform were investigated using methylene blue (MB) as a Raman probe dye. A quantitative study revealed excellent detection performance, with a limit of detection (LOD) of 42 pM for MB dye and a highly linear correlation between peak intensity and concentration (*R*^2^ ≥ 0.91). The SGBMNC platform also enabled the detection of toxic benzyl butyl phthalate with a sufficient LOD of 0.09 ppb (i.e., 280 pM). Therefore, we believe that the proposed methodology can be used for SERS assays of hazardous materials in practical fields.

## 1. Introduction

Plasmonic nanomaterials have been widely adopted in various applications, including optical sensing, photocatalysis, and nanophotonics, because of their ability to excite surface plasmons, which enhance light–matter interactions [1,2,3,4]. Electromagnetic fields are concentrated in nanoscale dielectric spaces (i.e., hotspots) in plasmonic arrays, resulting in localized surface plasmon resonance (LSPR) [5,6,7,8]. Such a phenomenon is particularly favorable for surface-enhanced Raman spectroscopy (SERS) analyses. SERS is a promising analytic tool to identify target analytes on the basis of inelastic scattering, which corresponds to unique vibrational and rotational modes in individual molecules [9]. Because single-molecule-level sensitivity can be achieved from elaborated nanostructures with high coherency and density, the hotspot engineering technique is a key factor in enhancing SERS performance. Despite devoted efforts, SERS has not yet been used as a commercial technique in practical assays because of the issues of cost, complexity, and reliability [5,7]. Therefore, an innovative approach is needed to develop a timesaving, convenient, and ultrasensitive SERS platform for on-site and point-of-care diagnoses of harmful materials encountered in daily life.

In the conventional development of SERS platforms, hotspot domains have been formed on surfaces or in the volumes of plasmonic nanostructures; thus, SERS signals are generated when analytes are placed in these restricted areas [10,11,12]. However, molecules with random diffusion can transfer and attach to any surface, including nonplasmonic or weak plasmonic regions, resulting in a degradation of the sensitivity and the limit-of-detection (LOD), especially at lower analyte concentrations [13,14]. Meanwhile, interior hotspots prepared by encapsulating analytes with metal layers have been introduced [14,15,16,17]. Under this concept, dye molecules adsorbed onto substrate surfaces can produce fingerprint spectra simultaneously during redox processes. Because the scale of the interior hotspots is similar to the size of molecules, these SERS platforms are highly desirable for detecting trace quantities of smaller analytes, at which intense electric fields (*E*-fields) are induced (i.e., the LSPR phenomenon).

Noble metals (e.g., Ag, Au, and Cu) have been commonly used as plasmonic materials because of their numerous free electrons. Although Ag exhibits the best plasmonic characteristics among the noble metals, its use remains challenging because of its high reactivity and susceptibility to surface oxidation [18,19]. To enhance the reliability and plasmonic activity of plasmonic materials, researchers have developed bimetallic nanocomposites (BMNCs) for Ag and Au [20,21,22,23]. Among various techniques, the galvanic replacement reaction (GRR) provides spontaneous transmetallation as a result of differences in the reduction potentials of two or more metal ions [22,23,24,25]. In general, when the reduction potential ($E^{o}$) of a sacrificial substrate is lower than that of other metal ions in a solution, their replacement occurs. Thus, for Ag–Au BMNCs (SGBMNCs), Ag ($E_{\mathrm{Ag}}^{o}$~0.8 V vs. standard hydrogen electrode (SHE)) is oxidized, whereas Au^3+^ or AuCl_4_^−^ ions (1.0 ≤ $E_{\mathrm{Au}}^{o}$ ≤ 1.5 V vs. SHE) are reduced [23,25,26]. Interestingly, hollow regions (e.g., interstitials and voids) are created in SGBMNCs during replacement because of the stoichiometric charge balance. Netzer et al. used GRR to fabricate Ag–Au bimetallic nanowires with a detection limit of 10 ppb for rhodamine 6G adsorbed onto an isolated nanowire [27]. Trace analyses of mercury ions using Ag–Au nanorods [28] and 1,4-benzenedithiol using Ag–Au nanocubes synthesized by ascorbic acid-associated co-reduction [29] have also been reported. Although SERS studies have been conducted for molecules adsorbed onto BMNCs after the completion of GRR, the interior hotspots have not been fully utilized.

In the present work, we prepared the SGBMNC SERS platform using chemical synthesis methods. The Ag nanoparticles (AgNPs) were synthesized on a plasma-treated cellulose acetate (pCA) paper substrate (AgNPs/pCA) via the Ag mirror reaction. Transmetallated SGBMNCs with interior hotspots were obtained through in situ galvanic replacement in the presence of analytes. We investigated the reproducibility, sensitivity, and LOD of the SGBMNC platform using methylene blue (MB) as a Raman probe dye. The function of interior hotspots was experimentally validated by comparing them to the platforms covered by droplets. To explore the feasibility of practical applications, the trace analysis of benzyl butyl phthalate (BBP) was also carried out using the proposed SERS platform.

## 2. Results

### 2.1. Sensing Strategy of the SGBMNC SERS Platform

A schematic of the fabrication procedures for the SGBMNC SERS platform using facile solution-based methods is presented in Appendix A. To enable the effective adsorption of AgNPs, the surface properties of cellulose acetate (CA) paper were investigated. Wettability plays a key role in the nucleation and growth of metal NPs by governing the access of metal ions to substrate surfaces. The nanoporous morphologies of bare CA papers stacked with the fibrous membranes were observed (Figure 1a). CA papers that exhibit hydrophobic character (*θ_c_*~100°) are well known to hinder the adsorption of metal ions onto their membranes [30,31]. As a result, a few AgNPs were formed after the Ag mirror reaction (Figure 1b). To address AgNPs on a substrate surface, we used plasma treatment to render the surface of CA papers hydrophilic. The CA (i.e., the organic ester of cellulose containing C, H, and O) was readily etched under the O_2_ plasma environment, resulting in an enhanced specific surface area (Figure 1c). Surface charges were induced on the pCA as a result of the dangling bonds of broken chains and the occurrence of hydroxyl groups, which enabled polar solvents (e.g., water) to closely approach the pCA surface [32]. In addition, negative charges, including hydroxyl groups, provided nucleation sites for Ag^+^ ions because of the attractive forces [33]. These features led to the successful formation of AgNPs with high density on the pCA (i.e., AgNPs/pCA) (Figure 1d).

The SERS behaviors of the AgNPs grown on bare and etched CA papers were characterized. The Ag deposition was performed by reacting Tollens’ reagent with glucose (0.5 M) under ambient conditions for 5 min. During the reactions, no external perturbations were observed. Afterward, 10 μL of 50 μM MB solution was dropped onto the substrate for characterization (Appendix A). A low-intensity spectrum of the MB-AgNPs/CA was recorded because of the absence of plasmonic nanostructures. By contrast, the spectrum of the MB-AgNPs/pCA showed an approximately 20-fold increase in signal intensity. As a result, we used the pCA papers as supporting substrates in subsequent experiments.

Compared with other techniques for synthesizing AgNPs (e.g., lithography, electrodeposition, and physical vapor deposition), the Ag mirror reaction method has the merits of facileness, cost-effectiveness, convenient morphological tunability, and compatibility with various substrate materials [34,35]. We, therefore, used this approach to prepare plasmonic AgNPs/pCA substrates. The electroless deposition is the result of a redox reaction between Tollens’ reagent and an aldehyde. In this study, diamminesilver(I) ions ([Ag(NH_3_)_2_]^+^) were used as a Tollens’ reagent; it was prepared via the reaction between AgNO_3_, NH_3_, and KOH (to adjust the pH). In the presence of an aldehyde (e.g., glucose), the [Ag(NH_3_)_2_]^+^ ions are reduced to Ag(0) atoms, leading to the nucleation and subsequent growth of AgNPs on the substrates. After the reaction, the oxidized gluconic acid can be bound to the synthesized AgNPs, stabilizing the AgNPs [36]. The as-mentioned reaction mechanism can be represented with the equation:(1)$$2{\left[\mathrm{Ag}{\left({\mathrm{NH}}_{3}\right)}_{2}\right]}^{+}+C_{5}H_{11}O_{5}\mathrm{CHO}+2{\mathrm{OH}}^{-}\to 2\mathrm{Ag}+C_{5}H_{11}O_{5}\mathrm{COOH}+H_{2}O+4{\mathrm{NH}}_{3}$$

The SGBMNCs embedded with analytes are prepared by reacting the AgNPs with solutions containing Au precursors and the target molecules (i.e., the GRR method). During the process, the AgNPs are generally oxidized to Ag^+^ ions, and the suspended AuCl_4_^−^ ions are reduced to solid Au(0) according to the differences in reduction potential ($E^{o}$) [26]:(2)$${\mathrm{Ag}}^{+}{}_{\left(aq\right)}\to \mathrm{Ag}\left(0\right)+1e^{-}\;(E_{\mathrm{Ag}}^{o}\sim 0.8\;V).$$
(3)$${\mathrm{AuCl}}_{4}{{}^{-}}_{\left(aq\right)}+3e^{-}\to \mathrm{Au}\left(0\right)+4{\mathrm{Cl}}^{-}{}_{\left(aq\right)}\;(E_{\mathrm{Au}}^{o}\sim 1.0\;V).$$

Interstitials are introduced into the AgNPs as a result of stoichiometric charge balancing, as described by the equation:(4)$$3\mathrm{Ag}\left(0\right)+{\mathrm{AuCl}}_{4}{{}^{-}}_{\left(aq\right)}\to 3{\mathrm{Ag}}^{+}{}_{\left(aq\right)}+\mathrm{Au}\left(0\right)+4{\mathrm{Cl}}^{-}{}_{\left(aq\right)}$$

A portion of the Ag^+^ ions might also be reduced to Ag(0) by accepting electrons [23]. These dynamic redox processes enable the growth of AgNPs with simultaneous encapsulation by Ag–Au bimetallic shells. Therefore, the analytes can be located in the developed narrow spaces in the SGBMNCs. Consequently, their SERS signals can be obtained with dramatically amplified intensities during the GRR reactions.

### 2.2. Morphological and Structural Properties

#### 2.2.1. AgNPs/pCA Substrate

The morphological properties of AgNPs/pCA with various Ag mirror reaction times were observed by field-emission scanning electron microscopy (FE-SEM) (Figure 2). At the early stage (i.e., 1 min), small AgNPs with high coverage were grown and distributed over the pCA. The AgNPs became larger with the increasing reaction time: Their diameters were 30 ± 3, 40 ± 10, 70 ± 15, and 80 ± 10 nm at 1, 3, 5, and 10 min, respectively. Notably, the interparticle distance between adjacent AgNPs strongly influences the SERS performance. Wide interparticle gaps were observed for the AgNPs grown at 1 and 3 min. The agglomerated AgNPs observed at 5 min provided sufficiently narrow gaps, whereas the AgNPs observed at 10 min were aggregated.

To investigate the plasmonic properties of the AgNPs/pCA substrates, we recorded the SERS spectra of MB (50 μM) on AgNPs/pCA substrates prepared at different Ag mirror reaction times (Appendix A). Overall, the MB-AgNPs/pCA substrates exhibited an excellent signal-to-noise ratio (SNR) compared with the MB-pCA substrates (i.e., 0 min). The most intense spectrum was recorded for the MB-AgNPs/pCA at 5 min, in good agreement with the structural characteristics. Accordingly, the optimum Ag mirror reaction time was determined to be 5 min. However, the lack of coherency in the scale, shape, and distribution of AgNPs limited the sensitive detection of analytes at lower concentrations. In the quantitative study, the MB signals were substantially degraded with the decreasing MB concentration, and the LOD of the AgNPs/pCA was directly determined to be 0.5 μM (Appendix A). Therefore, a novel strategy to further improve the plasmonic characteristics of AgNPs is required for effective SERS assays in practical fields.

#### 2.2.2. SGBMNC SERS Platform

The SGBMNCs were prepared using the GRR process. During the reactions, as described in Section 2.1, both the partial replacement of the Ag in AgNPs by Au and the growth of the entire domain with bimetallic shells proceeded simultaneously. The small-molecule analytes with metal ions were diffused to the AgNPs, which functionalized the interior hotspots. We analyzed the structural properties of the SGBMNC SERS platform as shown in FE-SEM images (Figure 3a–c). The scale of the SGBMNCs was gradually increased as the duration of the GRR process was lengthened. Though the transmetallic conversion between Ag and Au immediately occurs in the GRR, it is interesting that the Au deposition on AgNPs without internal hollow regions was observed at the early stage (i.e., 10 s) as shown in field-emission transmission electron microscopy (FE-TEM) images (Figure 3d,e). In the case of GRR, there are two possible routes: One is the replacement (i.e., the Kirkendall effect), and the other is the deposition. The dominant mechanism is determined by several factors, such as reaction temperature and reducing materials [22,24]. In our study, the gluconic acid and its salt on AgNPs acted as the reducing materials to the AuCl_4_^–^ ions rather than the stabilized Ag at the early stage [37]. With a longer process time, the Au reduction rate by Ag became dominant as confirmed in the FE-SEM images. In the face-centered cubic crystal system, the lattice constant of Ag (4.086 Å) is similar to that of Au (4.079 Å). Because of their negligible lattice mismatch, the Ag and Au are sufficiently miscible to enable the formation of bimetallic alloys [25]. Their amorphous crystallinity with a d-spacing of 0.24 nm was observed (corresponding to the (111) plane of Ag and Au) (Figure 3f). The atomic constituents of the SGBMNCs were investigated using elemental mapping (Figure 3g). The Ag was dominant because of the core nanostructures, whereas the partially replaced areas belonged to the Au. As a result, the SGBMNCs were successfully prepared with internal hotspots for sensitive SERS analyses.

### 2.3. Optimizing SERS Performance of the SGBMNC Platform

#### 2.3.1. Influence of Au Precursor Concentration

The concentration of the Au precursor (i.e., HAuCl_4_) is a key factor in controlling the rate of replacement of Ag with Au and the structural/optical properties of the SGBMNCs [23]. The replacement rate influences the density of crevices inside the AgNPs and the growth rate of bimetal shells. Therefore, the SERS performance of SGBMNC exhibits a trade-off relationship with the concentration of HAuCl_4_. To explore this phenomenon, MB-SGBMNC platforms were prepared with HAuCl_4_ concentrations ranging from 0.5 to 5.0 mM (Figure 4a). The signature features of MB molecules were observed in the Raman spectra of all the as-prepared substrates. Their intensity profiles at 1620 cm^–1^ are presented in Figure 4b. At the lower Au precursor concentrations (i.e., 0.5 and 1.0 mM), weak signals were generated because fewer hotspots were formed. With a higher replacement rate (i.e., 3.0 and 5.0 mM), the number of crevices inside the AgNPs increased but aggregation of the overall nanostructures was accelerated. In addition, the Au, which has a relatively weak intrinsic plasmonic property compared with that of Ag, occupied more sites, leading to a degradation of the SERS performance. Therefore, the optimum concentration of HAuCl_4_ solution was determined to be 2.0 mM.

#### 2.3.2. Effect of GRR Process Time

Another factor related to the SERS performance is the GRR process time. The spectral behaviors of MB-SGBMNC were investigated by monitoring the peak intensity at 1620 cm^–1^ at 5 s intervals (Figure 4c). Before the GRR, negligible background signals were obtained from the pristine AgNPs/pCA. At the early stage of GRR, the precursors and analytes were dynamically diffused and reacted with the substrates. The peak intensity in the spectrum of the MB-SGBMNC was markedly enhanced within 10 s to a level 29-fold higher than the intensity in the spectrum of MB-AgNPs/pCA. With a longer reaction, the plasmonic properties of SGBMNCs were substantially degraded. The signal enhancement was related to the coupling effect of interior hotspots with environmental hotspots [14]. The efficiency of the interior hotspots was related to the shell thickness as well as the skin depth, while that of the environmental hotspots (i.e., conventional hotspot regions between the AgNPs) was related to the gap distance. Under a long reaction time, the entire domain became much larger and aggregated, leading to the significant degradation of the *E*-field confinement in the environment hotspots. In the case of MB-AgNPs/pCA, the intensity did not substantially vary. Therefore, the optimum GRR time was selected as 10 s. The origin of the superior SERS behaviors of the SGBMNC platform was also investigated (Figure 4d). In both the case of MB-AgNPs/pCA and that of post-addition of MB to the SGBMNCs, the MB dye molecules were predominantly adsorbed onto the surfaces and produced weak SERS signals due to the random distributions. The remarkable enhancement was achieved from the MB-SGBMNC because the analytes diffused to the substrates became encapsulated to be functionalized as hotspot domains. To verify the hotspot locations (i.e., internal and environmental sites) where the MB dyes were present, the influence of washing on SERS intensity was investigated (Appendix A). Because MB molecules are highly water-soluble, the signals would be significantly decreased after washing when they are weakly entrapped by the SGBMNCs [16,38]. For the post-addition of MB to SGBMNC and MB-SGBMNC, washing was applied 10 times with deionized water, and the SERS signals were measured at each step. In the case of post-addition, most MB molecules were left from the surface of SGBMNC at the first washing, resulting in the peak intensity decreasing by 74.8%. In the case of MB-SGBMNC, the signal intensity was decreased by 30.3% after washing 10 times, which indicated that the MB molecules were successfully encapsulated by the SGBMNC. These facts directly indicate that the sufficient enhancement from the MB-SGBMNC was derived from the hotspots in the SGBMNCs.

### 2.4. SERS Activities of the SGBMNC Platform for MB Dyes

#### 2.4.1. Reproducibility

For accurate SERS sensing, every substrate should generate uniform signals under the same experimental conditions. The reproducibility of the SGBMNC platform was evaluated from 10 different substrates using an MB concentration of 1 μM (Figure 5a). The spectra show highly consistent features without deformations. To identify the MB dyes, we selected the peaks at 509, 1182, 1395, and 1620 cm^–1^ for the analytic criteria, corresponding to the in-plane bending of C–N–C bonds, in-plane bending of CH, stretching of C–N bonds, and stretching of C–C bonds, respectively [39]. The intensities at individual analytic peaks were found to be distributed within two standard deviations from the mean (Figure 5b). Consequently, the relative standard deviation (RSD) was calculated to be 3.28%, 5.87%, 3.47%, and 4.36% for the peaks at 509, 1182, 1395, and 1620 cm^–1^, respectively. We therefore used the developed SGBMNC platform with excellent reliability in further SERS analyses.

#### 2.4.2. Sensitivity

The sensitivity of the SGBMNC platform was investigated in the MB concentration range from 50 pM to 1 μM (Figure 6a). All spectra were collected at the GRR process time of 10 s. The spectra of MB-SGBMNC included the fingerprint spectrum of MB dye, distinguishing them from the spectra of the SGBMNC without MB dye (i.e., 0 M). The MB signals were observed until the MB concentration reached 50 pM. The relationship between the intensity and number of analytes was also investigated at four analytic peaks (Figure 6b). The SNR values of the peaks at 1182 and 1620 cm^–1^ were greater than those of the peaks at 509 and 1395 cm^–1^ because of the background noise. Highly linear quantitativeness was observed, with a correlation coefficient (*R*^2^) of 0.95, 0.98, 0.91, and 0.96 for the peaks at 509, 1182, 1395, and 1620 cm^–1^, respectively. The LOD of the MB-SGBMNC was evaluated using the equation [14]:(5)$$\mathrm{LOD}=\frac{3\times S_{b}}{m}$$
where *S*_b_ is the standard deviation of a blank sample (*n* = 3) and *m* is the slope of the calibration plot. Accordingly, the average LOD was determined to be 42 pM. Therefore, hazardous substances can potentially be monitored in a rapid and sensitive manner using the SGBMNC platform without any pretreatment or labeling process.

### 2.5. Label-Free SERS Detection of BBP Using the SGBMNC Platform

The SGBMNC SERS platform was used to detect toxic materials used in daily life. BBP, which is a commercial plasticizer, was selected as the target molecule because of its reprotoxic, carcinogenic, and endocrine-disrupting characteristics [40,41]. For comparison, BBP detection using the proposed SERS platforms without interior hotspots (i.e., BBP-AgNPs/pCA and post-addition of BBP to SGBMNC) were preliminarily investigated (Appendix A). The spectra for 10 ppm BBP were collected from individual substrates at the process time of 10 s. An intense BBP spectrum was provided by the highly sensitive interior hotspot, whereas weak SERS signals were generated from the BBP adsorbed only onto the substrate surface. For the AgNPs/pCA, the features vanished when the BBP concentration was 1 ppm (Appendix A).

The sensitivity of BBP-SGBMNC was investigated in the concentration range from 10^−1^ to 10^4^ ppb (Figure 7a). From all the tested platforms, the SERS spectra with the fingerprint peaks at 879, 1047, 1149, 1383, 1455, 1510, and 1596 cm^−1^ were clearly observed without significant shifts, demonstrating good agreement with other literature [42,43,44,45]. The intensity was gradually degraded as the BBP concentration decreased. To evaluate the concentration correlation, the intensities at 1389 and 1510 cm^−1^ were linearly regressed with a logarithm form (*n* = 3) (Figure 7b). Highly linear quantitative correlations were obtained, with *R*^2^ values of 0.99 and 0.98 at 1389 and 1510 cm^–1^, respectively. On the basis of Equation (3), the average LOD of BBP-SGBMNC was estimated to be 0.09 ppb (i.e., 0.28 nM). These results suggest that the new proposed methodology enabled the label-free detection of small toxic components with rapidity (10 s), ultrasensitivity, and reliability.

## 3. Discussion

The SERS platforms with interior hotspots demonstrated strong potential as an alternative to conventional sensing platforms. The nanostructures can be prepared using facile and rapid redox processes (e.g., GRR and electrochemical deposition) with convenient morphological tunability [14,15]. The GRR method provides spontaneous transmetallated nanostructures with SERS-active hollow regions into which analytes diffuse, whereas the electrochemical deposition approach results in molecules covered by metal layers formed via the reduction of metal ions by a reducing agent or by an applied potential. Sufficient *E*-fields become confined in these material-involved hotspots, leading to an enhancement of both the sensitivity and the LOD. In the present study, the proposed SGBMNC SERS platform prepared by the GRR method exhibits picomolar sensitivity for small dyes (Figure 6a).

The surface wettability of CA was modified under an O_2_ plasma environment. The charged surfaces attracted water-based solutions and provided nucleation sites for the Ag^+^ ions, resulting in a large population of AgNPs grown on the pCA substrates (Figure 1d) compared with the population of AgNPs on CA substrates (Figure 1b). The scale of AgNPs was continuously increased with the increasing Ag mirror reaction time (Figure 2). A gap distance appropriate for the LSPR coupling effect was achieved with the AgNPs/pCA at 5 min (Appendix A). To induce the formation of interior hotspots in the AgNPs/pCA and the diffusion of analytes simultaneously, the SGBMNCs were fabricated using the in situ GRR process with solutions containing HAuCl_4_ and MB dye. The plasmonic properties of SGBMNC could be tuned by adjusting the HAuCl_4_ concentration and the process time because of a trade-off relationship between the interior hotspot density, morphological aggregation, and intrinsic material properties; the optimum conditions were determined to be a HAuCl_4_ concentration of 2 mM and a process time of 10 s (Figure 4a–c). The contribution of interior hotspots to the SERS spectrum was also experimentally investigated. The MB dye molecules adsorbed onto the surfaces (i.e., post-addition of MB to SGBMNCs) produced a weak spectrum because of the traditional limitation, whereas those embedded into the internal regions exhibited drastically intensified signals (Figure 4d). These results are direct evidence of the presence of interior hotspots. Despite the active redox processes and molecular diffusion, the SGBMNC platform exhibited excellent reproducibility, with RSD values of less than 6% (Figure 5b).

Plastics that contain toxic plasticizers are commonly encountered in daily activities. Long-term exposure to them poses a substantial risk to human health. Therefore, extensive efforts have been devoted to detecting certain phthalate esters (e.g., BBP, diethyl hexyl phthalate, and dibutyl phthalate) to protect public safety. Conventional methods for phthalate analysis include mass spectrometry, high-performance liquid chromatography, and fluorescence spectrophotometry. Although these techniques provide sensitive detections, they require sophisticated equipment and labor-intensive procedures [42,45,46]. SERS has also been proposed as an attractive platform for the detection of phthalate plasticizers. However, because of the poor affinity of phthalate to metal substrates, labeling receptors (e.g., β-cyclodextrin) have been used in the SERS substrates for size-selective capture [44,47]. In the present work, the detection target was chosen as BBP, which is leached from products because of its noncovalent binding [42,48]. As a result, the SGBMNC platform enabled quantitative assays of BBP in 10 s without any labeling pretreatment.

In conclusion, the SGBMNC platform with interior hotspots provides useful insights into the development of novel hotspot engineering strategies for the rapid, sensitive, and reliable detection of toxic materials.

## 4. Materials and Methods

### 4.1. Materials

CA filter paper (diameter: 47 mm, pore size: 0.2 μm) was purchased from Hyundai Micro (Gyeongi, Korea). Silver nitrate (AgNO_3_), potassium hydroxide (KOH), d-(+)-glucose, gold (III) chloride trihydrate (HAuCl_4_·3H_2_O), MB, and BBP were purchased from Sigma-Aldrich (St. Louis, MO, USA). Ammonia solution (NH_3_, 28%) was purchased from Junsei (Tokyo, Japan). Ethyl alcohol anhydrous (99.9%) was obtained from Samchun (Gangnam, Korea). Deionized water (DIW) was obtained from Lian Corporation (Korea).

### 4.2. O_2_ Plasma Treatment of the CA Papers

The surface of CA paper was treated with O_2_ plasma using a 13.56 MHz reactive-ion-etching instrument (Vacuum Science). The plasma treatment was conducted at an O_2_ flow rate of 52 sccm and a working pressure of 2.5 × 10^−2^ Torr for 120 s.

### 4.3. Tollens’ Reagent Preparation

A Tollens’ reagent was used as the Ag metal precursor for electroless Ag deposition. The solution was prepared by mixing aqueous solutions of AgNO_3_ (0.5 M, 5 mL) and KOH (0.8 M, 660 μL), which produced thick brown precipitates. NH_3_ solution was then added to the mixture dropwise until the precipitates were dissolved completely. The resultant solution was immediately used to synthesize AgNPs.

### 4.4. Ag Mirror Reaction for the Synthesis of AgNPs/pCA

A pCA substrate was dipped in glucose solution (0.5 M) for 5 s and then carefully transferred into a Petri dish containing the Tollens’ reagent. The Ag mirror reaction was then conducted for 5 min (unless stated otherwise). Afterward, the substrate was carefully washed with DIW and then dried under ambient conditions. The substrate was subsequently cut into small pieces (0.5 cm × 0.5 cm) for further processing.

### 4.5. GRR Process for the Fabrication of the SGBMNCs Platform and SERS Measurement

A customized deposition well was prepared for the GRR process and simultaneous SERS measurement. In the deposition well, solutions of HAuCl_4_ and the target analytes (total volume of 400 µL) at desired concentrations were carefully added. An Ocean Optics portable probe spectrometer system (UQEPRO-Raman) was used to monitor the SERS signals. A laser at 785 nm with an optical power of 40 mW was incident to the well. The SERS signals were recorded with an acquisition time of 5 s.

### 4.6. Characterization

The morphologies of the SGBMNCs were characterized by FE-SEM (JEOL JSM-6700F) and FE-TEM (JEM-2100F, Jeol, Tokyo, Japan).

## Figures and Tables

**Figure 1 ijms-23-11741-f001:**
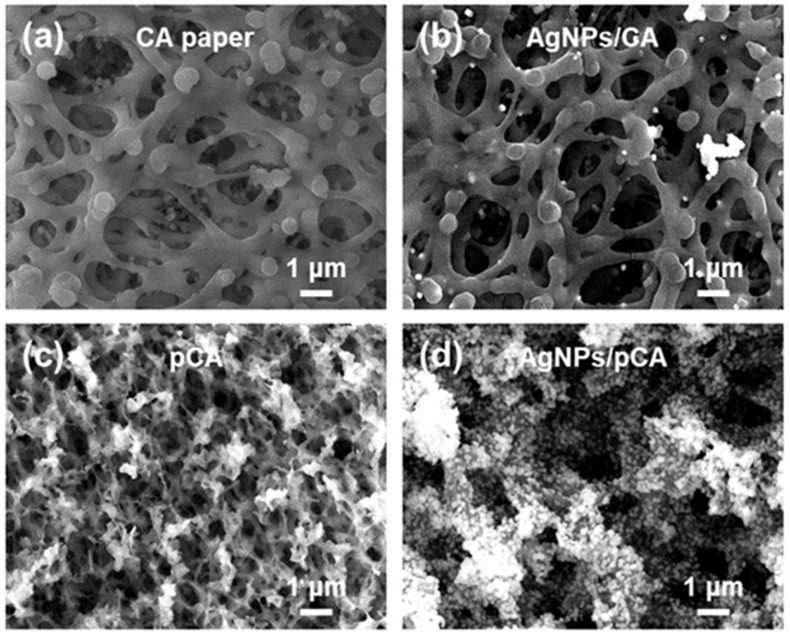
Influence of surface wettability on the synthesis of AgNPs. FE-SEM images of (**a**,**b**) the CA and (**c**,**d**) the pCA substrates (**a**,**c**) before and (**b**,**d**) after the Ag mirror reaction.

**Figure 2 ijms-23-11741-f002:**
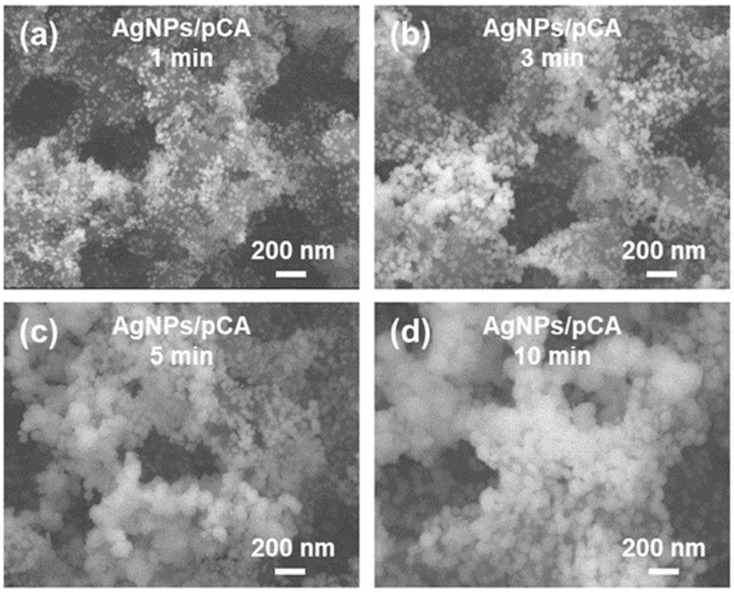
Morphological properties of the AgNPs/pCA substrate. FE-SEM images of the AgNPs/pCA at Ag mirror reaction times of (**a**) 1, (**b**) 3, (**c**) 5, and (**d**) 10 min.

**Figure 3 ijms-23-11741-f003:**
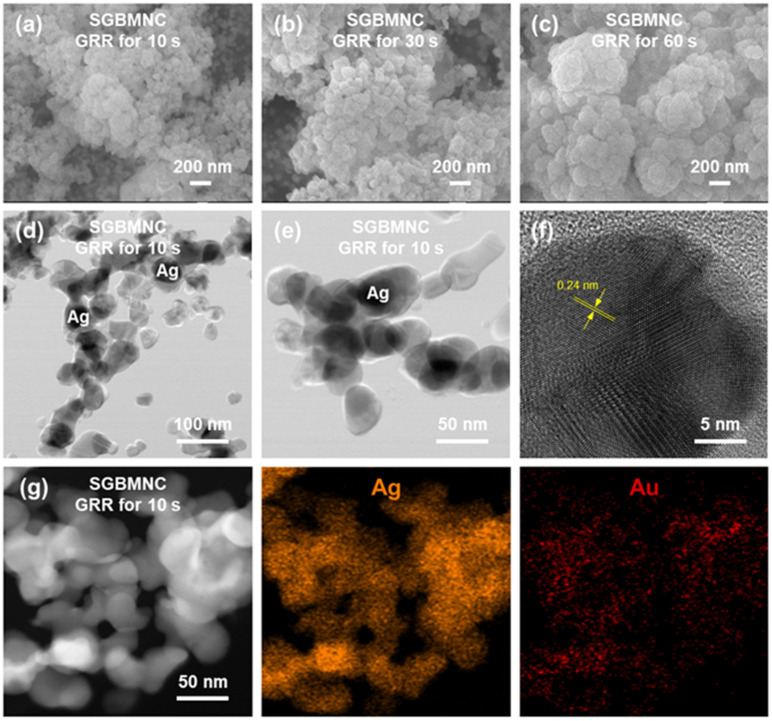
Morphological and crystalline properties of the SGBMNC platform. FE-SEM images of the SGBMNC at GRR times of (**a**) 10, (**b**) 30, and (**c**) 60 s. (**d**,**e**) FE-TEM images, (**f**) HR-TEM image, and (**g**) elemental mapping results of the SGBMNC.

**Figure 4 ijms-23-11741-f004:**
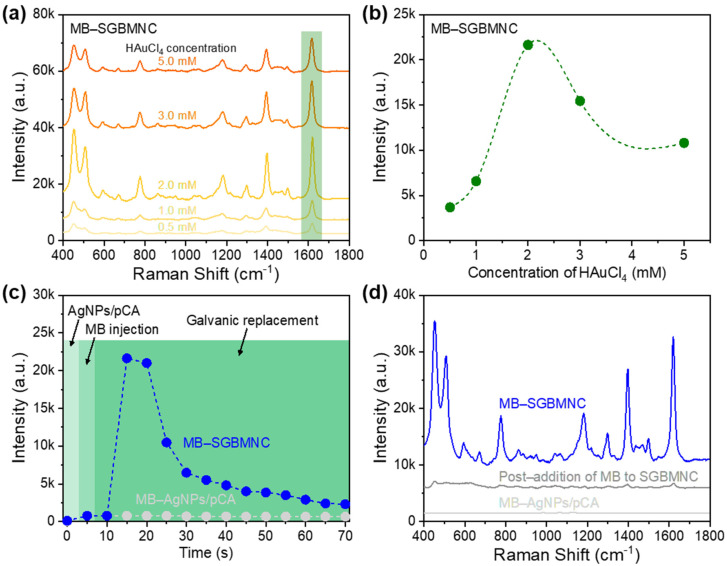
Optimization of the performance of the SGBMNC SERS platform. (**a**) Influence of the Au precursor concentration and (**b**) the corresponding intensity profile. (**c**) Real-time monitoring for the SERS spectra of the MB-SGBMNC and MB-AgNPs/pCA and (**d**) a comparison of the SERS platforms with and without the interior hotspots.

**Figure 5 ijms-23-11741-f005:**
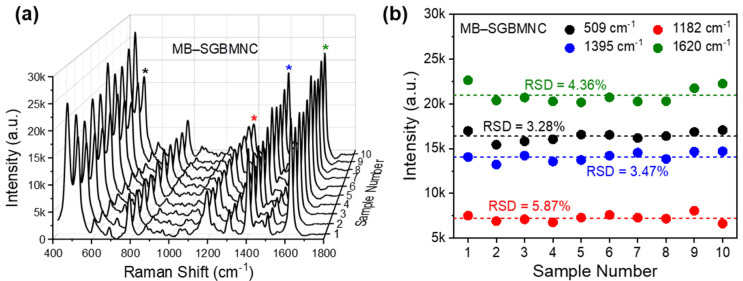
Reproducibility of the SGBMNC SERS platform for MB dyes. (**a**) SERS spectra obtained from 10 different platforms and (**b**) their intensity profiles at 509 (*), 1182 (*), 1395 (*), and 1620 (*) cm^−1^.

**Figure 6 ijms-23-11741-f006:**
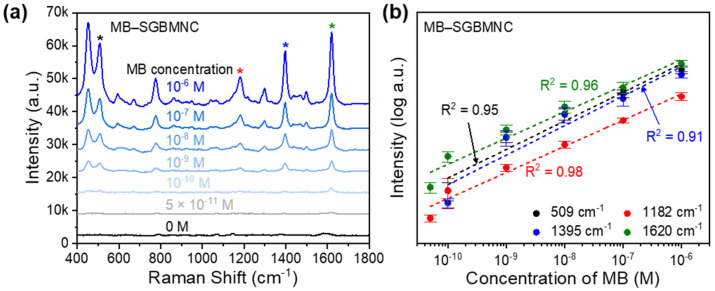
Sensitivity and LOD of the SGBMNC for MB dye. (**a**) Quantitative SERS analysis with MB concentrations from 50 pM to 1 µM and (**b**) their corresponding intensity variations at the analytic peaks at 509 (*), 1182 (*), 1395 (*), and 1620 (*) cm^−1^. The error bars correspond to the standard deviation (*n* = 3).

**Figure 7 ijms-23-11741-f007:**
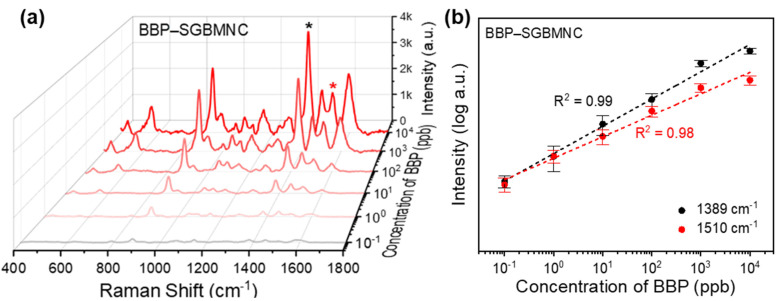
SERS assays of the BBP. (**a**) Quantitative SERS analysis with BBP concentrations from 0.1 to 1000 ppb and (**b**) their corresponding intensity variations at the analytic peaks at 1389 (*) and 1510 (*) cm^−1^. The error bars correspond to the standard deviation (*n* = 3).

## Data Availability

Not applicable.

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
