# Peer review of "Interior Hotspot Engineering in Ag–Au Bimetallic Nanocomposites by In Situ Galvanic Replacement Reaction for Rapid and Sensitive Surface-Enhanced Raman Spectroscopy Detection"

_ijms, 2022, doi:10.3390/ijms231911741_

Round 1

Reviewer 1 Report

This manuscript describes a relatively straight-forward method of generating enhanced SERS signals from bimetallic Au/Ag nanoparticles for the detection of methylene blue and other possible Raman active, toxic molecules.  The authors demonstrate that the detection limit is dramatically increased in the case where the molecules are first deposited on the Ag NP and then the galvanic displacement reaction with Au occurs.  When the molecules are added subsequent to the replacement, the strong enhancement is not observed.  The data are compelling, and the method may be useful for practical SERS detection of toxic molecules at relevant concentrations.  The manuscript might be improved if the authors could address the points below:

1)    What is the evidence that the molecules are encapsulated in the structure during the galvanic displacement?  The data clearly show that depositing the molecules post replacement does not give the same enhancement, but is it possible the moleculesjust cannot access the most active sites post fabrication?  Perhaps the wording/description could be improved here.

2)    On the bottom of page 5, the authors suggest that the Au first deposits on top of the AgNP but later changed to a mechanism where the Ag is oxidized and replaced, but the evidence for this is not clear.  (The authors also use the term “amorphous crystallinity here which is not clear.) The Ag and Au distributions are only shown at one time, and it is not clear what shows the mechanistic change suggested. 

3)    The decrease in the SERS response at long deposition times (4c) is suggested to be caused by the incorporation of higher amounts of gold, with a reduced SERS signal compared to Ag.  What amount/ratio of Ag/Au is needed? Does the distribution matter?  It is clear that the signal decreases, but the explanation for the signal decrease is not strong.

4)    It is not clear what this statement at on pg 4 means: “Overall, the MB-AgNPs/pCA substrates exhibited an excellent signal-to-noise ratio (SNR) compared with the pristine AgNPs/pCA substrates”

5)    The evidence for the proposed bimetallic shell structure formed upon galvanic replacement is not completely clear.  The compositional mapping may provide some evidence, but it is not clear. 

Overall, the evidence seems compelling and the method seems to provide a feasible method for fabrication of SERS sensing materials with low detection limits for these molecules and the results should be of use to the SERS community.

Author Response

<Comment 1> This manuscript describes a relatively straight-forward method of generating enhanced SERS signals from bimetallic Au/Ag nanoparticles for the detection of methylene blue and other possible Raman active, toxic molecules.  The authors demonstrate that the detection limit is dramatically increased in the case where the molecules are first deposited on the Ag NP and then the galvanic displacement reaction with Au occurs.  When the molecules are added subsequent to the replacement, the strong enhancement is not observed.  The data are compelling, and the method may be useful for practical SERS detection of toxic molecules at relevant concentrations. The manuscript might be improved if the authors could address the points below:

<Response> We appreciate the reviewer for carefully reading our manuscript and giving us valuable comments. We have addressed the concerns during the revision. Please check the revised manuscript.

<Comment 1> What is the evidence that the molecules are encapsulated in the structure during the galvanic displacement? The data clearly show that depositing the molecules post replacement does not give the same enhancement, but is it possible the molecules just cannot access the most active sites post fabrication? Perhaps the wording/description could be improved here.

<Response> The replacement of one metal with another is performed according to the catalyst, temperature, and different reduction potentials. In the absence of catalyst and high-temperature reactions, Ag ( ~ 0.8 V vs SHE) is readily replaced by Au (1.0 ≤  ≤ 1.5 V vs SHE). In this case, three phenomena were observed; i) generating hollow regions in Ag nanostructures, ii) increasing a ratio of Au/Ag, and iii) expanding the entire domain. However, in this study, the AgNPs synthesized by Ag mirror reaction had residual gluconic acids on their surfaces. The AuCl4 ions suspended in solutions preferentially reacted with gluconic acids (i.e., deposition) and with AgNPs (i.e., replacement). Therefore, the Au-coated AgNPs were observed in the GRR process for 10 s, followed by the conventional replacement for further reaction times (Figures 3 and 4c). In both cases, small analytes diffused to the substrates were positioned in the SGBMNC.

To experimentally support the formation of interior hotspots, the SERS intensity variations of the MB-SGBMNC and post-addition of MB to SGBMNC along with several washing times were investigated as shown in the figure below. Because MB molecules are highly water-soluble, the signals would be significantly decreased after washing when they are weakly entrapped by the SGBMNCs [a,b]. During ten times washing of substrates with deionized water, the SERS signals were measured at each step. In the case of post-addition, most MB molecules were left from the surface of SGBMNC at the first washing, resulting in the peak intensity decreasing by 74.8%. In the case of MB-SGBMNC, the signal intensity was decreased by 30.3% after ten times washing, which indicated that the MB molecules were successfully encapsulated by the SGBMNC.

We added the influence of washing on signal variations to experimentally prove the formation of interior hotspots with high stability in the revised manuscript and supplementary material.

  1. Ansah, I.B.; Kim, S.; Yang, J.-Y.; Mun, C.; Jung, H.S.; Lee, S.; Kim, D.-H.; Kim, S.-H.; Park, S.-G. In Situ Electrodeposition of Gold Nanostructures in 3D Ultra-Thin Hydrogel Skins for Direct Molecular Detection in Complex Mixtures with High Sensitivity. Laser Photonics Rev. 2021, 15, 2100316.
  2. Ma, L.; Chen, Y.-L.; Yang, D.-J.; Ding, S.-J.; Xiong, L.; Qin, P.-L.; Chen, X.-B. Gap-Dependent Plasmon Coupling in Au/AgAu Hybrids for Improved SERS Performance. J. Phys. Chem. C 2020, 124, 25473−25479.

Figure S4. Signal variations of the post-addition of MB to SGBMNC and MB-SGBMNC during ten times washing.

<Comment 2> On the bottom of page 5, the authors suggest that the Au first deposits on top of the AgNP but later changed to a mechanism where the Ag is oxidized and replaced, but the evidence for this is not clear.  (The authors also use the term “amorphous crystallinity here which is not clear.) The Ag and Au distributions are only shown at one time, and it is not clear what shows the mechanistic change suggested.

<Response> For the Au-Ag bimetallic composites, there are two routes in the GRR process; i) replacement of Ag by Au and ii) Au deposition on Ag nanostructures. According to the reaction temperature and reducing materials, the dominant mechanism is determined.

In the case of i) with the absence of any other catalyst, three Ag atoms are replaced by one Au atom due to their stoichiometric ratio [c], leading to the creation of interior hollow regions and the increment of the Au/Ag ratio. Furthermore, the oxidized Ag ions also had the opportunity to be reduced back to solid Ag, and these oxidations and reductions of Ag were repeated during the dynamic GRR process, resulting in the rapid growth of the entire domain. In the case of ii), other reducing materials are involved in the Au reduction. Because the Au precursors do not take the ions from the Ag, it is observed that the conformal Au deposition on the Ag nanostructures and slow growth of the entire domain, as compared to the case of i).

In this study, the AgNPs were prepared through the Ag mirror reaction and the resultant gluconic acid residues were partially present on the AgNPs. The reaction of AuCl4 ions with the gluconic acids made the Au deposition on AgNPs. As shown in FE-SEM and TEM images, the SGBMNC with the GRR for 10 s exhibited that iii) the total structures were not varied much, iv) Au atoms were reduced on the AgNPs, and v) there were no hollow regions. Because Au (4.078 Å) and Ag (4.086 Å) have good agreement in lattice constant, the amorphous crystallinity of synthesized SGBMNC was observed. When the GRR time was increased, the SGBMNC was remarkedly grown due to the reaction of AuCl4 ions with AgNPs. This transmetallation process between AuCl4 ions and Ag nanostructures can provide the Au-Ag nanocomposites/alloys with high thermodynamic stability [d-g]. Therefore, both the deposition and replacement were mentioned to describe the GRR phenomenon of the SGBMNC in this work.

[c] Ngamaroonchote, A.; Karn-orachai, K. Bimetallic Au–Ag on a Patterned Substrate Derived from Discarded Blu-ray Discs: Simple, Inexpensive, Stable, and Reproducible Surface-Enhanced Raman Scattering Substrates. Langmuir 2021, 37, 7392–7404.

[d] Xia, X.; Wang, Y.; Ruditskiy, A.; Xia, Y. 25th Anniversary Article: Galvanic Replacement: A Simple and Versatile Route to Hollow Nanostructures with Tunable and Well-Controlled Properties. Adv. Mater. 2013, 25, 6313−6333.

[e] Chee, S.W.; Tan, S.F.; Baraissov, Z.; Bosman, M.; Mirsaidov, U. Direct observation of the nanoscale Kirkendall effect during galvanic replacement reactions. Nat. Commun. 2017, 8, 1224.

[f] Netzer, N.L.; Qiu, C.; Zhang, Y.; Lin, C.; Zhang, L.; Fong, H.; Jiang, C. Gold−silver bimetallic porous nanowires for sur-face-enhanced Raman scattering. Chem. Commun. 2011, 47, 9606−9608.

[g] Lin, G.; Dong, W.; Wang, C.; Lu, W. Mechanistic study on galvanic replacement reaction and synthesis of Ag-Au alloy nanoboxes with good surface-enhanced Raman scattering activity to detect melamine. Sens. Actuators B Chem. 2018, 263, 274−280.

<Comment 3> The decrease in the SERS response at long deposition times (4c) is suggested to be caused by the incorporation of higher amounts of gold, with a reduced SERS signal compared to Ag. What amount/ratio of Ag/Au is needed? Does the distribution matter? It is clear that the signal decreases, but the explanation for the signal decrease is not strong.

<Response> The SERS activities of such hotspot designs are related to the coupling effect of interior hotspots with environmental hotspots. The E-fields are induced in the molecules inside the noble metal nanostructures (i.e., interior hotspots) and their efficiencies are based on the shell thickness as well as the skin depth. The E-fields are also confined in the spaces between the AgNPs (i.e., environmental hotspots) by following the traditional trends. These fields played an important role to enhance the penetration of incident lights and the intensity of scattered lights. The influence of the coupling effect on the SERS performance was experimentally and theoretically investigated in our previous study [14].

In this work, the entire domain became much larger and finally aggregated when the GRR process became longer. Consequently, the E-fields confined in the environmental hotspots were significantly degraded. We described this phenomenon in the revised manuscript.

<Comment 4> It is not clear what this statement at on pg 4 means: “Overall, the MB-AgNPs/pCA substrates exhibited an excellent signal-to-noise ratio (SNR) compared with the pristine AgNPs/pCA substrates”.

<Response> The signal-to-noise ratio (SNR) compares the Raman intensity of target molecules with that of background signals (i.e., substrate, solvent, etc). In Figure S3, the SNR of plasmonic AgNPs to pCA substrate was demonstrated. The term of pristine AgNPs/pCA substrates indicated the MB-AgNPs(0 min)/pCA. According to the reviewer’s comment, this term could be confusing to the audience, and thus we changed the sentence as shown in below in the revised manuscript.

“Overall, the MB-AgNPs/pCA substrates exhibited an excellent signal-to-noise ratio (SNR) compared with the MB-pCA substrates.”

<Comment 5> The evidence for the proposed bimetallic shell structure formed upon galvanic replacement is not completely clear.  The compositional mapping may provide some evidence, but it is not clear.

<Response> The SGBMNC prepared by deposition and replacement during the GRR process was discussed in this manuscript. The role of gluconic acids on AgNPs was addressed in the reviewer’s comment 2, leading to the Au deposition at an early stage and the replacement of Ag by Au in further stages. For comparison, another investigation of ours will be introduced in the upcoming paper; the GRR process for the Ag nanopillars. The Ag nanopillars were prepared by using vacuum techniques and thus their surface contained no chemical residues. When the AuCl4 ions were injected, therefore, it was found that the hollow regions (i.e., interior hotspots) were rapidly formed. The STEM image of Ag nanopillars under the GRR process for 10 s was represented below. Notably, the residues on the sacrificial materials significantly influence the redox process and thus comprehension of the mechanism is required.

Figure A. STEM image of Ag nanopillars under the GRR process for 10 s.

We also additionally verified the functions of the interior and environmental hotspots in the reviewer’s comment 1. The MB encapsulated by the SGBMNC was confirmed by slow signal loss during the washing with deionized water. In contrast, the signal was rapidly decreased in the MB post-added to the surface of SGBMNC due to the high water solubility.

Reviewer 2 Report

The paper "Interior hotspot engineering in Ag–Au bimetallic nanocomposites by in situ galvanic replacement reaction for rapid and sensitive surface-enhanced Raman spectroscopy detection" investigates the formation of hot-spots within Ag-Au bimetallic nanoparticles for SERS detection. 

The paper is well written and the authors investigate many aspects regarding the synthesis of the nanoparticles making sound conclusions. 

I recommend the paper to be published as it is

Author Response

We appreciate the reviewer for the positive evaluation of our manuscript.

Reviewer 3 Report

Manuscript Number: ijms-1880972

Title: Interior hotspot engineering in Ag–Au bimetallic nanocomposites by in situ galvanic replacement reaction for rapid and sensitive surface-enhanced Raman spectroscopy detection

Recommendation: Major Revisions
Reviewer comments:
    Technical Comments to the Author:

The authors developed a Ag–Au bimetallic nanocomposite (SGBMNC) SERS platform with interior hotspots through facile chemical syntheses. This is a very interesting thing. However, the part content of this work still need to be clarified. Therefore, it is recommended to issue after major revisions.

Remarks to the Author:

1. "Engineering of interior hotspots" is a new concept, but the author's sample type seems to be seen in other peers. Therefore, is it appropriate to use such a concept here?

2. In the introduction, "Despite devoted efforts, SERS has not yet been used as a commercial technique in practical assays because of the issues of cost, complexity, and reliability." Such a conclusion should be strongly supported by literature. Similar to this sentence, "However, molecules with random diffusion can transfer and attach to any surface, including nonplasmonic or weak plasmonic regions, resulting in a degradation of the sensitivity and the limit-of-detection (LOD), especially at lower analyte concentrations." and so on.

there is little background knowledge about the preparation method, detection and application technology of SERS substrate.

3. In line 172, I did not find the Section 3.1 mentioned in the manuscript.

4. If possible, some important and recent reports on SERS substrate preparation (e.g., Opt. Commun. 510(2022) 127921, Nanophotonics11.1 (2022): 33-44, and so on) should be added to show clear background.

5.The analysis and explanation of Figure 4d are too few to explain that remarkably enhanced the plasmonic activity compared to the conventional SERS platforms without the internal hotspots.

Author Response

<General comment> The authors developed a Ag–Au bimetallic nanocomposite (SGBMNC) SERS platform with interior hotspots through facile chemical syntheses. This is a very interesting thing. However, the part content of this work still need to be clarified. Therefore, it is recommended to issue after major revisions.

<Response> We appreciate the referee for giving us the opportunity to enhance our manuscript. We have addressed the concerns during the revision. Please check the revised manuscript.

<Comment 1> "Engineering of interior hotspots" is a new concept, but the author's sample type seems to be seen in other peers. Therefore, is it appropriate to use such a concept here?

<Response> We appreciate this comment for careful consideration of our fabrication protocols as compared with others. Conventional SERS approaches employ the made-and-dispersed technique to make analytes adsorbed on the surfaces. Although such a method is generally effective to detect molecules at high concentrations, random distributions and out-of-hotspot adsorption inhibit SERS assays at low molecular concentrations, especially, in chemical and biomedical fields. Therefore, the design of interior hotspots has been developed. In the chemistry field, the SERS analyses of analytes encapsulated by reduced metal layers have been investigated by using the potential-driven electrochemical deposition [a-c] and GRR processes [d-f]. However, the traditional limitations are still challenging. Recently, Lim et al. prepared the Au shell-gap-Au core nanoparticles through the in situ Au reduction and obtained the SERS signals from organic components when they were positioned in the gap areas [g].

Accordingly, the in situ GRR process in the presence of analytes was carried out to form the bimetallic nanocomposites in this work. The MB-SGBMNC demonstrated excellent SERS activities in sensitive, rapid, label-free, and reproducible manners. Moreover, we additionally verified the utilization and stability of interior hotspots in SGBMNC by tracing the intensity variations after washing with deionized water, as answered in the reviewer’s comment 5.

Therefore, we believe that the phrase “engineering of interior hotspots” fits as an appropriate terminology to describe the strategy of plasmonic nanostructures embedded with molecules, including the in situ GRR technique used in this work.

  1. Zhai, Y.; Zhu, Z.; Zhou, S.; Zhu, C.; Dong, S. Recent advances in spectroelectrochemistry. Nanoscale 2018, 10, 3089–3111.
  2. Wu, D.-Y.; Li, J.-F.; Ren, B.; Tian, Z.-Q. Electrochemical surface-enhanced Raman spectroscopy of nanostructures. Chem. Soc. Rev. 2008, 37, 1025−1041.
  3. Farling, C.G.; Stackaruk, M.C.; Pye, C.C.; Brosseau, C.L. Fabrication of high quality electrochemical SERS (EC-SERS) substrates using physical vapour deposition. Phys. Chem. Chem. Phys. 2021, 23, 20065–20072.
  4. Yang, Y.; Liu, J.; Fu, Z.-W.; Qin, D. Galvanic Replacement-Free Deposition of Au on Ag for Core−Shell Nanocubes with Enhanced Chemical Stability and SERS Activity. J. Am. Chem. Soc. 2014, 136, 8153−8156.
  5. Wang, L.; Patskovsky, S.; Gauthier-Soumis, B.; Meunier, M. Porous Au–Ag Nanoparticles from Galvanic Replacement Applied as Single-Particle SERS Probe for Quantitative Monitoring. Small 2022, 18, 2105209.
  6. Lee, T.; Jung, D.; Wi, J.-S.; Lim, H.; Lee, J.-J. Surfactant-free galvanic replacement for synthesis of raspberry-like silver nanostructure pattern with multiple hot-spots as sensitive and reproducible SERS substrates. Appl. Surf. Sci. 2020, 505, 144548.
  7. Lim, D.-K.; Jeon, K.-S.; Hwang, J.-H.; Kim, H.; Kwon, S.; Suh, Y.D.; Nam, J.-M. Highly uniform and reproducible sur-face-enhanced Raman scattering from DNA-tailorable nanoparticles with 1-nm interior gap. Nat. Nanotechnol. 2011, 6, 452−460.

<Comment 2> In the introduction, "Despite devoted efforts, SERS has not yet been used as a commercial technique in practical assays because of the issues of cost, complexity, and reliability." Such a conclusion should be strongly supported by literature. Similar to this sentence, "However, molecules with random diffusion can transfer and attach to any surface, including nonplasmonic or weak plasmonic regions, resulting in a degradation of the sensitivity and the limit-of-detection (LOD), especially at lower analyte concentrations." and so on.

<Response> According to the reviewer’s comment, we added the references to the statements as shown in below.

“Despite devoted efforts, SERS has not yet been used as a commercial technique in practical assays because of the issues of cost, complexity, and reliability [5,7].”

“However, molecules with random diffusion can transfer and attach to any surface, including nonplasmonic or weak plasmonic regions, resulting in a degradation of the sensitivity and the limit-of-detection (LOD), especially at lower analyte concentrations [13,14].”

We also added reference 13 as below.

  1. Lee, S.H.; Kim, S.; Yang, J.-Y.; Mun, C.; Lee, S.; Kim, S.-H.; Park, S.-G. Hydrogel-Assisted 3D Volumetric Hotspot for Sensitive Detection by Surface-Enhanced Raman Spectroscopy. Int. J. Mol. Sci. 2022, 23, 1004.

there is little background knowledge about the preparation method, detection and application technology of SERS substrate.

<Comment 3> In line 172, I did not find the Section 3.1 mentioned in the manuscript.

<Response> We corrected it from “Section 3.1” to “Section 2.1” in the revised manuscript.

<Comment 4> If possible, some important and recent reports on SERS substrate preparation (e.g., Opt. Commun. 510(2022) 127921, Nanophotonics11.1 (2022): 33-44, and so on) should be added to show clear background.

<Response> According to the reviewer’s suggestion, more references were added to explain the recent works in SERS research fields to the audiences as below.

  1. Guo, L.; Cao, H.; Cao, L.; Yang, Y.; Wang, M. SERS study of wheat leaves substrates with two different structures. Opt. Commun. 2022, 510, 127921.
  2. Zhang, C.; Li, Z.; Qiu, S.; Lu, W.; Shao, M.; Ji, C.; Wang, G.; Zhao, X.; Yu, J.; Li, Z. Highly ordered arrays of hat-shaped hierarchical nanostructures with different curvatures for sensitive SERS and plasmon-driven catalysis. Nanophotonics 2022, 11, 33−44.

<Comment 5> The analysis and explanation of Figure 4d are too few to explain that remarkably enhanced the plasmonic activity compared to the conventional SERS platforms without the internal hotspots.

<Response> According to the reviewer’s comment, we improved the description for Figure 4d by investigating the influence of washing on SERS signals to experimentally verify where the MB molecules were present. The SERS intensity variations of the MB-SGBMNC and post-addition of MB to SGBMNC along with several washing times were carried out as shown in the figure below. Because MB molecules are highly water-soluble, the signals would be significantly decreased after washing when they are weakly entrapped by the SGBMNCs [h,i]. During ten times washing of substrates with deionized water, the SERS signals were measured at each step. In the case of post-addition, most MB molecules were left from the surface of SGBMNC at the first washing, resulting in the peak intensity decreasing by 74.8%. In the case of MB-SGBMNC, the signal intensity was decreased by 30.3% after ten times washing, which indicated that the MB molecules were successfully encapsulated by the SGBMNC.

We added the influence of washing on signal variations to experimentally prove the formation of interior hotspots with high stability in the revised manuscript and supplementary material.

  1. Ansah, I.B.; Kim, S.; Yang, J.-Y.; Mun, C.; Jung, H.S.; Lee, S.; Kim, D.-H.; Kim, S.-H.; Park, S.-G. In Situ Electrodeposition of Gold Nanostructures in 3D Ultra-Thin Hydrogel Skins for Direct Molecular Detection in Complex Mixtures with High Sensitivity. Laser Photonics Rev. 2021, 15, 2100316.
  2. Ma, L.; Chen, Y.-L.; Yang, D.-J.; Ding, S.-J.; Xiong, L.; Qin, P.-L.; Chen, X.-B. Gap-Dependent Plasmon Coupling in Au/AgAu Hybrids for Improved SERS Performance. J. Phys. Chem. C 2020, 124, 25473−25479.

Figure S4. Signal variations of the post-addition of MB to SGBMNC and MB-SGBMNC during ten times washing.

Round 2

Reviewer 3 Report

I have no any questions.